# Asymptotics of smoothed Wasserstein distances in the small noise regime

**Yunzi Ding**[1]    **Jonathan Niles-Weed**[2]
[1]Courant Institute of Mathematical Sciences, NYU
[2]Courant Institute of Mathematical Sciences and the Center for Data Science, NYU
yunziding@gmail.com
jnw@cims.nyu.edu

## Abstract

We study the behavior of the Wasserstein-2 distance between discrete measures $\mu$ and $\nu$ in $\mathbb{R}^d$ when both measures are smoothed by small amounts of Gaussian noise. This procedure, known as *Gaussian-smoothed optimal transport*, has recently attracted attention as a statistically attractive alternative to the unregularized Wasserstein distance. We give precise bounds on the approximation properties of this proposal in the small noise regime, and establish the existence of a phase transition: we show that, if the optimal transport plan from $\mu$ to $\nu$ is unique and a perfect matching, there exists a critical threshold such that the difference between $W_2(\mu, \nu)$ and the Gaussian-smoothed OT distance $W_2(\mu * \mathcal{N}_\sigma, \nu * \mathcal{N}_\sigma)$ scales like $\exp(-c/\sigma^2)$ for $\sigma$ below the threshold, and scales like $\sigma$ above it. These results establish that for $\sigma$ sufficiently small, the smoothed Wasserstein distance approximates the unregularized distance exponentially well.

## 1 Introduction: optimal transport

Optimal Transport (OT) has seen a recent surge of applications in machine learning, in areas such as generative modeling [2, 16], image processing [13, 27, 31], and domain adaptation [7, 8]. A natural statistical question raised by these applications is to estimate the OT distances with samples. These distances, known as the Wasserstein distances, are defined by

$$W_p^p(\mu, \nu) = \inf_{\pi \in \Pi(\mu,\nu)} \int \|x - y\|^p d\pi(x, y),$$

where $\Pi(\mu, \nu)$ denotes the set of joint measures with marginals $\mu$ and $\nu$, known as *transport plans*. It is well known that plug-in estimators for this quantity, obtained by replacing $\mu$ and $\nu$ with empirical measures consisting of i.i.d. samples, have performance in high dimensions, with rates of convergence typically of order $n^{-2/d}$ [3, 11, 12, 15, 22] when $d > 2p$. Moreover, minimax lower bounds show that this curse of dimensionality is unavoidable in general [25, 32].

The existence of the curse of dimensionality for OT has led to a series of proposals to obtain better rates of convergence by imposing additional structural assumptions—such as latent low-dimensionality [25] or smoothness [24, 33]—or by replacing $W_p$ by a better-behaved surrogate, such as an entropy-regularized version with much better statistical and computational properties [1, 9, 17, 23, 28].

A particularly intriguing option, developed by [18], consists in *smoothing* the Wasserstein distance by adding Gaussian noise. The following result shows the statistical benefits of this approach.

**Proposition 1.1** ([21]). *For $d > 1$ and $\sigma > 0$, denote by $\mathcal{N}_\sigma$ the centered Gaussian measure on $\mathbb{R}^d$ with covariance $\sigma^2 I_d$. For any compactly supported probability measure $\mu$ in $\mathbb{R}^d$, let $x_1, x_2, \ldots, x_n$*

*be i.i.d. samples from μ, and define the empirical measure*

$$\hat{\mu}_n = \frac{1}{n}\sum_{i=1}^{n}\delta(x_i).$$

*Then there exists a constant $c = c(\mu, \sigma, d)$ such that*

$$\mathbb{E}W_2(\hat{\mu}_n * \mathcal{N}_\sigma, \mu * \mathcal{N}_\sigma) \leq cn^{-1/2}.$$

[18] call this framework *Gaussian-smoothed optimal transport* (GOT), and follow up work has shown that it possesses significant statistical benefits, with fast rates of convergence and clean limit laws [19, 20, 38].

To leverage the beneficial properties of the GOT framework, it is necessary to understand how well the smoothed distance $W_2(\mu * \mathcal{N}_\sigma, \nu * \mathcal{N}_\sigma)$ approximates the standard Wasserstein distance $W_2(\mu, \nu)$. An application of the triangle inequality [18, Lemma 1] shows that

$$|W_2(\mu, \nu) - W_2(\mu * \mathcal{N}_\sigma, \nu * \mathcal{N}_\sigma)| \lesssim \sigma. \tag{1}$$

Indeed, the triangle inequality implies $|W_2(\mu*\mathcal{N}_\sigma, \nu*\mathcal{N}_\sigma) - W_2(\mu, \nu)| \leq W_2(\mu, \mu*\mathcal{N}_\sigma) + W_2(\nu, \nu*\mathcal{N}_\sigma)$ and the latter two terms are of order at most $\sigma$. In general, this upper bound is unimprovable, as we show below. On the other hand, it can also be very loose: if $\mu$ is a translation of $\nu$, then $W_2(\mu * \mathcal{N}_\sigma, \nu * \mathcal{N}_\sigma) = W_2(\mu, \nu)$ for all $\sigma \geq 0$. These examples raise a natural question: how well does $W_2(\mu * \mathcal{N}_\sigma, \nu * \mathcal{N}_\sigma)$ approximate $W_2(\mu, \nu)$ when $\sigma$ is small, and how does the answer to this question depend on the measures $\mu$ and $\nu$?

The main goal if this paper is to give a sharp answer to this question for *finitely supported* measures. We focus on the finite support case for two reasons. First, when $\mu$ and $\nu$ are finitely supported, $\mu * \mathcal{N}_\sigma$ and $\nu * \mathcal{N}_\sigma$ are each finite mixtures of Gaussians, and the behavior of Wasserstein distances for such measures is a topic of active research [6, 10]. Second, as our results indicate, the behavior of this quantity for finitely supported measures is unexpectedly rich, with a sharp dichotomy in rates depending on the structure of the optimal transport plan between $\mu$ and $\nu$: we show that when the *unique* optimal transport plan between $\mu$ and $\nu$ is a *perfect matching*, then there exist positive $\sigma_*$ and $c$ such that

$$0 \leq W_2(\mu, \nu) - W_2(\mu * \mathcal{N}_\sigma, \nu * \mathcal{N}_\sigma) \lesssim e^{-c/\sigma^2} \quad \forall \sigma \in (0, \sigma_*).$$

In other words, for sufficiently small $\sigma$, the GOT distance approximates the standard $W_2$ distance exponentially well, substantially sharpening (1). More strikingly, we establish the existence of a phase transition: for $\sigma < \sigma_*$, the gap is exponentially small, whereas for $\sigma > \sigma_*$, the gap scales linearly. By contrast, if the optimal transport plan between $\mu$ and $\nu$ is not unique or is not a perfect matching, then no phase transition appears: the upper bound of (1) is tight even in a neighborhood of $\sigma = 0$.

Our work provides a precise understanding on how GOT resembles vanilla OT in the vanishing noise ($\sigma \downarrow 0$) regime. These results complement those recently obtained by [5] in the large noise regime, who show that if $\mu$ and $\nu$ have $n$ matching moments, $n \geq 1$, then $W_2(\mu * \mathcal{N}_\sigma, \nu * \mathcal{N}_\sigma) = O(\sigma^{-n})$ as $\sigma \to \infty$. Along with results in [5], our work completes the limiting picture of the Euclidean heat semigroup acting on atomic measures under the Wasserstein distance. All the relevant rates are presented in Table 1.

We note that our work leaves open the question of characterizing the rates for non-atomic measures. It is possible to show that, for general measures, there are measures exhibiting polynomial rates intermediate between $\sigma$ and $e^{-c/\sigma^2}$; however, these rates appear to depend delicately on the geometry of the measures and their support. Giving a full characterization of the rate for general probability measures is an attractive open question.

## 2 Preliminaries and main results

We are concerned with the optimal transport problem between discrete measures

$$\mu = \sum_{i=1}^{k}\alpha_i\delta(x_i), \quad \nu = \sum_{j=1}^{\ell}\beta_j\delta(y_j)$$

| Regime | Condition | $\lim(W_2(\mu * \mathcal{N}_\sigma, \nu * \mathcal{N}_\sigma))$ | Rate | Reference |
|---|---|---|---|---|
| $\sigma \downarrow 0$ | Unique perfect matching | $W_2(\mu, \nu)$ | $e^{-c/\sigma^2}$ | Theorem 4.1 |
| $\sigma \downarrow 0$ | No unique perfect matching | $W_2(\mu, \nu)$ | $\sigma$ | Theorem 4.4 |
| $\sigma \uparrow \infty$ | $\mu$ and $\nu$ agree up to $n$th moment | $0$ | $\sigma^{-n}$ | [5] |

Table 1: Limiting behavior of $W_2(\mu * \mathcal{N}_\sigma, \nu * \mathcal{N}_\sigma)$ for atomic measures $\mu$ and $\nu$.

in the space $\mathbb{R}^d$, equipped with the squared Euclidean cost function $c(x, y) = \|x - y\|^2$. Here $\{\alpha_i\}_{i=1}^k$ and $\{\beta_j\}_{j=1}^\ell$ are positive numbers such that $\sum_{i=1}^k \alpha_i = \sum_{j=1}^\ell \beta_j = 1$, and the sets $\mathcal{X} := \{x_i\}$ and $\mathcal{Y} := \{y_j\}$ consist of distinct elements of $\mathbb{R}^d$. (Note that we do not require that $\mathcal{X} \cap \mathcal{Y} = \emptyset$.) Explicitly, we may write

$$W_2^2(\mu, \nu) = \inf_{\pi \in \Pi(\mu, \nu)} \int \|x - y\|^2 d\pi(x, y) = \min_{\pi \in \Pi(\mu, \nu)} \sum_{i \in [k], j \in [\ell]} \|x_i - y_j\|^2 \pi(x_i, y_j). \quad (2)$$

We call a minimizer in (2) an *optimal coupling*.

We shall show that the behavior of the quantity $W_2(\mu * \mathcal{N}_\sigma, \nu * \mathcal{N}_\sigma)$ as $\sigma \to 0$ depends strongly on the structure of the optimal couplings between $\mu$ and $\nu$. We rely on the following definition.

**Definition 2.1.** *The measures $\mu$ and $\nu$ possess a* unique perfect matching *if there exists a unique solution $\pi^*$ to (2), and if the support of this unique solution is a perfect matching, i.e., the set $\{x \in \mathcal{X}, y \in \mathcal{Y} : \pi^*(x, y) > 0\}$ is a bijection between $\mathcal{X}$ and $\mathcal{Y}$.*

Our main results (Theorems 4.1 and 5.1) show that the GOT distance approximates the Wasserstein distance exponentially well for small $\sigma$ *if and only if* the measures possess a unique perfect matching. To obtain these bounds, we show that if $\sigma$ is small and $\mu$ and $\nu$ possess a unique perfect matching, then the optimal plan for $\mu$ and $\nu$ is also approximately optimal, in an appropriate sense, for the convolved measures $\mu * \mathcal{N}_\sigma$ and $\nu * \mathcal{N}_\sigma$. By contrast, if $\mu$ and $\nu$ do not possess a unique perfect matching, then we explicitly exhibit an alternate coupling between $\mu * \mathcal{N}_\sigma$ and $\nu * \mathcal{N}_\sigma$ with significantly smaller cost, therefore showing that $W_2(\mu * \mathcal{N}_\sigma, \nu * \mathcal{N}_\sigma)$ is smaller than $W_2(\mu, \nu)$ by an amount that scales linearly in $\sigma$.

To identify the range of $\sigma$ for which the exponential error bound holds, we introduce a robust version of optimality for $\pi^*$ in the perfect matching case. This definition, *strong cyclical monotonicity*, extends the classical cyclical monotonicity criterion from optimal transport [see, e.g. 37], and captures how sensitive the optimal plan is to perturbations of the source and target measure. We show that this notion is closely related to the strong convexity and smoothness of the dual optimal solutions to the optimal transport problem, known as "potentials." The strong convexity of these potentials has previously been explored in computational and statistical contexts [26, 36], but to our knowledge its connection to the stability of discrete optimal tranport plans is new.

## 3 Strong cyclical monotonicity

In this section, we consider transport plans in the form of a perfect matching between $\{x_i\}$ and $\{y_i\}$. By relabeling the points, we may assume without loss of generality that the optimal transport plan between $\mu$ and $\nu$ is the matching given by

$$\Gamma = \{(x_1, y_1), (x_2, y_2), \dots, (x_k, y_k)\}.$$

In this section, we develop a robust notion of optimality for $\Gamma$. This notion is based on a strengthening of the classic optimality condition for optimal transport, based on cyclical monotonicity. We recall the following definition.

**Definition 3.1** (See, e.g., 30). *A set $S \subseteq \mathbb{R}^d \times \mathbb{R}^d$ is* cyclically monotone *if for any $(a_1, b_1), \dots, (a_n, b_n) \in S$, we have*

$$\sum_{i=1}^n \|a_i - b_i\|^2 \leq \sum_{i=1}^n \|a_i - b_{i+1}\|^2,$$

*where we set $b_{n+1} := b_1$.*

The significance of this notion is the following fundamental result.

**Theorem 3.2** (See 37, Theorem 5.10)**.** *If $\pi \in \Pi(\mu, \nu)$ has cyclically monotone support, then it is an optimal transport plan between $\mu$ and $\nu$.*

Our main definition strengthens this characterization by requiring the inequalities in the definition of cyclical monotonicity to be strict.

**Definition 3.3.** *We say $f : [k] \times [k] \to \mathbb{R}_{\geq 0}$ is a positive residual function on $[k]$, if $f(i, i) = 0$, $f(i, j) > 0$ for $i \neq j$, and $f(i, j) = f(j, i)$ for all $i, j \in [k]$.*

**Definition 3.4** (Strong cyclical monotonicity)**.** *For a positive residual function $f$ on $[k]$, we say that $\Gamma$ is $f$-strongly cyclically monotone, if for any $1 \leq n \leq k$ and distinct $\tau(1), \tau(2), \ldots, \tau(n) \in [k]$ (with the convention $\tau(n + 1) = \tau(1)$), we have*

$$\sum_{i=1}^{n} \|x_{\tau(i)} - y_{\tau(i)}\|^2 \leq \sum_{i=1}^{n} \|x_{\tau(i)} - y_{\tau(i+1)}\|^2 - \sum_{i=1}^{n} f(\tau(i), \tau(i+1)),$$

*or equivalently,*

$$\sum_{i=1}^{n} \langle x_{\tau(i)}, y_{\tau(i)} - y_{\tau(i+1)} \rangle \geq \sum_{i=1}^{n} f(\tau(i), \tau(i+1)).$$

Strong cyclical monotonicity indicates that the optimal plan with support $\Gamma$ is superior to any other plan by a positive margin in its transport cost. The importance of Definition 3.4 is that is equivalent to robustness of the optimality of $\Gamma$ under small perturbations of the points $\{x_i\}$ and $\{y_i\}$. To make this connection precise, we make the following definition.

**Definition 3.5.** *For $\epsilon \geq 0$, we say $\Gamma$ is $\epsilon$-robust, if for any distinct $\tau(1), \tau(2), \ldots, \tau(n) \in [k]$, and any $\alpha_{\tau(1)}, \alpha_{\tau(2)}, \ldots, \alpha_{\tau(n)} \in \mathbb{R}^d$ such that*

$$\max_{i} \|\alpha_{\tau(i)}\| \leq \epsilon,$$

*there holds*

$$\sum_{i=1}^{n} \|x_{\tau(i)} - y_{\tau(i)}\|^2 \leq \sum_{i=1}^{n} \|(x_{\tau(i)} + \alpha_{\tau(i)}) - (y_{\tau(i+1)} + \alpha_{\tau(i+1)})\|^2.$$

*We write*

$$R(\Gamma) := \sup \{ \epsilon \geq 0 \ : \ \Gamma \text{ is } \epsilon\text{-robust} \}.$$

The quantity $R(\Gamma)$, which we call "robustness of optimality," captures the behavior of the optimal plan when the supports of $\mu$ and $\nu$ are slightly perturbed. As the following proposition indicates, robustness in this sense is equivalent to strong cyclical monotonicity.

**Proposition 3.6.** *$\Gamma$ is strongly cyclically monotone if and only if $R(\Gamma) > 0$.*

Proposition 3.6 may be viewed as a robust analogue to Theorem 3.2—if $\pi \in \Pi(\mu, \nu)$ has *strong* cyclically monotone support, then it is a robustly optimal transport plan between $\mu$ and $\nu$, in the sense that it remains an optimal plan even when $\mu$ and $\nu$ are corrupted by noise. As we establish in Section 4, this observation is central to the analysis of GOT.

## 3.1 Implementability and explicit bounds on $R(\Gamma)$

Despite the mathematical simplicity of Definition 3.4, it is not clear how to verify it for a particular set $\Gamma$, nor how to establish that this property holds for an optimal plan between $\mu$ and $\nu$. To this end, we propose another condition, *strong implementability*, which is equivalent to strong cyclical monotonicity but is more amenable to analysis. We also show that both conditions are equivalent to $\mu$ and $\nu$ possessing a unique perfect matching.

[29] introduced the notion *implementability* and established it as an equivalent condition of cyclical monotonicity. In parallel to the results in [29], we also introduce the following stronger condition of implementability.

**Definition 3.7** (Strong implementability). *For a positive residual function $f$ on $[k]$, we say that $\Gamma$ is $f$-strongly implementable, if there exists a potential function $\varphi$, such that for any $i, j \in [k]$, we have*

$$\langle x_i, y_i - y_j \rangle \geq \varphi(y_i) - \varphi(y_j) + f(i, j).$$

Analogous to the equivalence result in [29], we show that strong cyclical monotonicity and strong implementability are both equivalent to the uniqueness and optimality of $\Gamma$.

**Proposition 3.8.** *The following three statements are equivalent:*

*(i) $\Gamma$ is $f$-strongly cyclically monotone for some $f$;*

*(ii) $\Gamma$ is $f$-strongly implementable;*

*(iii) $\Gamma$ is the unique optimal transport plan from $\{x_i\}$ to $\{y_i\}$.*

**Remark 3.9.** *We can imply from the direction (i) to (iii) of Proposition 3.8 that, if the directed bipartite graph with vertex set $\{x_1, x_2, \ldots, x_k, y_1, y_2, \ldots, y_k\}$ and arcs between $x_i$ and $y_j$ with weight $\|x_i - y_j\|^2$ does not possess an alternating cycle of zero total cost, then $\Gamma$ is the unique optimal transport plan from $\{x_i\}$ to $\{y_i\}$. One may use this sufficient condition to verify uniqueness of an optimal transport plan $\Gamma$ in practice.*

The equivalence in Proposition 3.8 holds for any positive residual function $f$; however, in the context of optimal transport with the squared Euclidean cost, it is most natural to focus on the quadratic case. The positive residual function constructed in the equivalence between (iii) and (i) in Proposition 3.8 is of the form $f(i, j) = \frac{\lambda}{2}\|y_i - y_j\|^2$ for some $\lambda > 0$, in which case the implementability condition reads

$$\langle x_i, y_i - y_j \rangle \geq \varphi(y_i) - \varphi(y_j) + \frac{\lambda}{2}\|y_i - y_j\|^2.$$

Quadratic residual functions are closely connected to convex analysis and to the theory of optimal transport. This condition is equivalent to the existence of a $\lambda$-strongly convex potential $\varphi$ satisfying $\nabla\varphi(y_i) = x_i$ for all $i \in [k]$ [35], or, equivalently, the existence of a Lipschitz *Brenier map* from $\mu$ to $\nu$ [4]. The regularity of Brenier maps is a deep question in analysis [see, e.g. 14]. Proposition 3.8 establishes that, in the finite-support case, this question is equivalent to the uniqueness of the optimal transport plan for $\mu$ and $\nu$.

More generally, we have the following theorem characterizing the properties of strongly implementable plans with residual functions of quadratic type.

**Theorem 3.10.** *The following conditions are equivalent:*

*(i) For some $0 \leq \alpha < \beta$, there exists a potential function $\varphi : \mathbb{R}^d \to \mathbb{R}^d$ which is $\alpha$-strongly convex and $\beta$-smooth, such that $x_i = \nabla\varphi(y_i)$ for all $i \in [k]$.*

*(ii) $\Gamma$ is strongly implementable for*

$$f(i, j) := \frac{1}{2(\beta - \alpha)} \left( \|x_i - x_j\|^2 + \alpha\beta\|y_i - y_j\|^2 - 2\alpha\langle y_i - y_j, x_i - x_j \rangle \right), \quad (3)$$

*or equivalently, there exists $\{\tilde{\varphi}(y_i)\}_{i=1}^{k} \subset \mathbb{R}^d$, such that for all $i, j \in [k]$ $(i \neq j)$,*

$$\langle x_i, y_i - y_j \rangle \geq \tilde{\varphi}(y_i) - \tilde{\varphi}(y_j)$$
$$+ \frac{1}{2(\beta - \alpha)} \left( \|x_i - x_j\|^2 + \alpha\beta\|y_i - y_j\|^2 - 2\alpha\langle y_i - y_j, x_i - x_j \rangle \right) \quad (4)$$

*Proof.* This is a direct application of Theorem 4 in [35]. The condition (i) in Theorem 3.10 is equivalent to the set $\{(y_i, x_i, \varphi(y_i))\}$ being $\mathcal{F}_{\alpha,\beta}$-interpolable in Definition 2 of [35], and the condition (ii) is equivalent to equation (4) in Theorem 4 of [35]. $\square$

Theorem 3.10 also formally encompasses the choice $\beta = +\infty$ when $\varphi$ is strongly convex but not smooth, in which case (3) reads

$$f(i, j) := \frac{\alpha}{2}\|y_i - y_j\|^2, \quad (5)$$

which recovers the positive residual function used in the proof of Proposition 3.6.

**Remark 3.11.** *We should emphasize that the $f$ defined in Theorem 3.10 is indeed a positive residual function given $\alpha < \beta$, since Cauchy-Schwartz gives*

$$2\alpha\langle y_i - y_j, x_i - x_j \rangle \leq \|x_i - x_j\|^2 + \alpha^2\|y_i - y_j\|^2 < \|x_i - x_j\|^2 + \alpha\beta\|y_i - y_j\|^2.$$

As a direct consequence of the direction (ii) to (i) in Theorem 3.10, if $\Gamma$ is strongly implementable for a positive residual function $f$ which is quadratic in $y_i - y_j$ and $x_i - x_j$, there exists a smooth and strongly convex potential function verifying implementability.

**Corollary 3.12.** *Suppose $\Gamma$ is strongly implementable for*

$$f(i,j) = \frac{1}{2}\left(\lambda_{xx}\|x_i - x_j\|^2 + \lambda_{yy}\|y_i - y_j\|^2 - 2\lambda_{xy}\langle y_i - y_j, x_i - x_j \rangle\right)$$

*where $\lambda_{xx}, \lambda_{xy}$ and $\lambda_{yy}$ are nonnegative numbers which satisfy $\lambda_{xy}^2 + \lambda_{xy} = \lambda_{xx}\lambda_{yy}$. Then there exists a potential function $\varphi : \mathbb{R}^d \to \mathbb{R}^d$ which is $\frac{\lambda_{xy}}{\lambda_{xx}}$-strongly convex and $\frac{\lambda_{yy}}{\lambda_{xy}}$-smooth, such that $x_i = \nabla\varphi(y_i)$ for all $i \in [k]$.*

By the Smith–Knott optimality criterion for optimal transport [34], the conclusion that $x_i = \nabla\varphi(y_i)$ for all $i \in [k]$ is equivalent to the potential function $\varphi$ solving the following *dual* version of (2):

$$\inf_\phi \sum_{i=1}^{k} \alpha_i \phi^*(x_i) + \sum_{j=1}^{\ell} \beta_j \phi(y_j)\,, \tag{6}$$

where $\phi^*$ denotes the Legendre conjugate.

Finally, we show that the above characterizations give rise to lower bounds on $R(\Gamma)$, which are easy to compute in $O(k^2)$ time. The following bound gives a quantitative link between robustness of optimality and the residual function $f$.

**Proposition 3.13.** *Suppose $\Gamma$ is strongly implementable for a positive residual function $f$. Then $\Gamma$ is $\epsilon$-robust for*

$$\epsilon \leq \frac{1}{2}\inf_{i \neq j} \frac{f(i,j)}{\|x_i - x_j\| + \|y_i - y_j\|}. \tag{7}$$

*This implies that*

$$R(\Gamma) \geq \frac{1}{2}\inf_{i \neq j} \frac{f(i,j)}{\|x_i - x_j\| + \|y_i - y_j\|}.$$

By combining this bound with Theorem 3.10, we obtain a simple lower bound when $f$ is of quadratic type.

**Proposition 3.14.** *When the equivalence in Theorem 3.10 holds, $\Gamma$ is $\epsilon$-robust for*

$$\epsilon \leq \frac{1}{2}\inf_{i \neq j} \frac{\max\left\{\frac{1}{\beta}\|x_i - x_j\|^2, \alpha\|y_i - y_j\|^2\right\}}{\|x_i - x_j\| + \|y_i - y_j\|}. \tag{8}$$

*This implies that*

$$R(\Gamma) \geq \frac{1}{2}\inf_{i \neq j} \frac{\max\left\{\frac{1}{\beta}\|x_i - x_j\|^2, \alpha\|y_i - y_j\|^2\right\}}{\|x_i - x_j\| + \|y_i - y_j\|}.$$

**Remark 3.15.** *When condition (i) in Theorem 3.10 holds, $\alpha$-strong convexity and $\beta$-smoothness implies*

$$\frac{1}{\beta}\|x_i - x_j\| \leq \|y_i - y_j\| \leq \frac{1}{\alpha}\|x_i - x_j\|.$$

*Thus the condition (8) may be replaced by the bound*

$$\epsilon \leq \frac{1}{2}\inf_{i \neq j}\max\left\{\frac{\alpha}{1+\beta}\|x_i - x_j\|, \frac{\alpha}{\beta(1+\alpha)}\|y_i - y_j\|\right\}, \tag{9}$$

*which is more interpretable. We should emphasize that, to check the $\epsilon$-robustness of $T$ with either (8) or (9) requires prior knowledge on the parameters $\alpha$ and $\beta$, which are inherent to the optimal transport plan $\Gamma$.*

# 4 Exponential rates for unique perfect matchings

Our main results show that the robustness of optimality $R(\Gamma)$ controls the gap between $W_2(\mu, \nu)$ and $W_2(\mu * \mathcal{N}_\sigma, \nu * \mathcal{N}_\sigma)$.

**Theorem 4.1.** *If $\sigma_* = R(\Gamma) > 0$, then for $\sigma \in (0, \sigma_*)$,*

$$W_2(\mu, \nu) - W_2(\mu * \mathcal{N}_\sigma, \nu * \mathcal{N}_\sigma) \lesssim \sqrt{\sigma_* \sigma} e^{-\sigma_*^2/4\sigma^2}.$$

The proof depends on the following simple lemma, which shows that robustness to optimality implies that the Wasserstein distance is *unchanged* when $\mu$ and $\nu$ are corrupted by small noise.

**Lemma 4.2.** *If $\sigma_* = R(\Gamma) > 0$, then for any measure $Q$ in $\mathbb{R}^d$ supported on $B(0, \sigma_*)$,*

$$W_2(\mu, \nu) = W_2(\mu * Q, \nu * Q).$$

Proofs of these results appear in the supplementary material.

In the regime where $\sigma$ does not exceed $R(\Gamma)$, the above theorem tells that the GOT distance is an excellent approximation of the OT distance. Our second main result is a converse to that statement, showing that if $\sigma$ goes beyond $R(\Gamma)$, the loss $W_2(\mu, \nu) - W_2(\mu * \mathcal{N}_\sigma, \nu * \mathcal{N}_\sigma)$ is bounded below by a linear function of $\sigma$. We start with the following proposition, which quantifies a "violation of cyclical monotonicity" under possibly large perturbations in the sources and targets.

**Proposition 4.3.** *If $\Gamma$ is an optimal transport plan, for any $M \geq 0$, denote*

$$G(M) := \sup \left\{ \sum_{i=1}^n \|x_{\tau(i)} - y_{\tau(i)}\|^2 - \sum_{i=1}^n \|(x_{\tau(i)} + \alpha_{\tau(i)}) - (y_{\tau(i+1)} + \alpha_{\tau(i+1)})\|^2 : \|\alpha_{\tau(i)}\| \leq M \right\}$$

*Then $G(M)$ is a concave function of $M$ for $M \in [0, +\infty)$.*

Note that $G(M)$ vanishes for $M < \sigma_*$. The next theorem shows that as long as $G(M)$ is not negligible for $M \gtrsim \sigma_*$, the approximation loss for $\sigma \geq \sigma_*$ is linear in $\sigma$.

**Theorem 4.4.** *If $\sigma_* = R(\Gamma) > 0$, then*

$$W_2^2(\mu, \nu) - W_2^2(\mu * \mathcal{N}_\sigma, \nu * \mathcal{N}_\sigma) \gtrsim \sup_{M > \sigma_*} e^{-M^2/\sigma^2} G(M).$$

*Here $G(M)$ is defined as in Proposition 4.3. In particular, if $G(3\sigma_*) \geq c_0 \sigma_*$ for an absolute constant $c_0 > 0$, then there exists a constant $C = C(c_0) > 0$ such that for $\sigma \in (0, 2\sigma_*)$,*

$$W_2^2(\mu, \nu) - W_2^2(\mu * \mathcal{N}_\sigma, \nu * \mathcal{N}_\sigma) \geq C\sigma e^{-\sigma_*^2/\sigma^2}.$$

The proof of Theorem 4.1, Lemma 4.2, Proposition 4.3, and Theorem 4.4 can be found in the supplementary material.

# 5 Beyond perfect matchings

In the case that $R(\Gamma) = 0$, or equivalently by Proposition 3.8 and Proposition 3.6 that the optimal transport map between $\mu$ and $\nu$ is not a perfect matching, Theorems 4.1 and 4.4 are not applicable. In this situation, we are able to show that the approximation error is linear, even in a neighborhood of zero. In fact, this holds whenever there exists an optimal transport plan between $\mu$ and $\nu$ which is not a perfect matching.

To analyze this case, we return to the setting of general discrete measures:

$$\mu = \sum_{i=1}^m \alpha_i \delta(x_i), \quad \nu = \sum_{j=1}^n \beta_j \delta(y_j). \tag{10}$$

**Theorem 5.1.** *Let $\mu$ and $\nu$ be as in (10). If $\mu$ and $\nu$ do not possess a unique perfect matching, then there exists $c_0 > 0$ such that for $\sigma \in (0, c_0)$,*

$$W_2^2(\mu, \nu) - W_2^2(\mu * \mathcal{N}_\sigma, \nu * \mathcal{N}_\sigma) \gtrsim \sigma.$$

Theorem 5.1 tells that, unless the optimal transport plan between $\mu$ and $\nu$ is unique and a perfect matching, the loss from approximating the OT distance with the GOT distance is at least linear in $\sigma$. We derive Theorem 5.1 from the following lemma, which shows that Theorem 5.1 holds in the special case where $\mu$ is a single point mass, and $\nu$ is uniform on two points.

**Lemma 5.2.** *Let $x, y_1$ and $y_2$ be different points in $\mathbb{R}^d$. For $\mu_0 := \delta(x)$ and $\nu_0 := \frac{1}{2}\delta(y_1) + \frac{1}{2}\delta(y_2)$, there exists $c_0 > 0$, such that for $\sigma \in (0, c_0)$, we have*

$$W_2^2(\mu_0, \nu_0) - W_2^2(\mu_0 * \mathcal{N}_\sigma, \nu_0 * \mathcal{N}_\sigma) \gtrsim \sigma. \tag{11}$$

We obtain the full strength of Theorem 5.1 by reducing to the special case of Lemma 5.2 on a particular subset of the support of $\mu$ and $\nu$. Full details appear in the supplementary material.

# 6   Numerical example

In this section, we present a numerical example to demonstrate different regimes of the rate $W_2(\mu, \nu) - W_2(\mu * \mathcal{N}_\sigma, \nu * \mathcal{N}_\sigma)$, in respect of Theorem 4.1 and Theorem 4.4. For the sake of clarity, we consider atomic measures $\mu$ and $\nu$ both defined on $\mathbb{R}^2$. One of the simplest cases where a coupling $\Gamma$ has $R(\Gamma) = 0$ is

$$\mu = \frac{1}{2}\left[\delta((-1, -1)) + \delta((1, 1))\right],$$

$$\nu = \frac{1}{2}\left[\delta((-1, 1)) + \delta((1, -1))\right]$$

It is easy to see that the optimal transport plan from $\mu$ to $\nu$ is not unique, which is also a consequence of Proposition 3.8, Proposition 3.6 and the fact that $R(\Gamma) = 0$ for the map

$$\Gamma = \{((-1, -1), (-1, 1)), ((1, 1), (1, -1))\}$$

that achieves the optimal cost. We also consider the family of perturbed measures

$$\mu(p) = \frac{1}{2}\left[\delta((-1, -1 + p)) + \delta((1, 1 - p))\right], \ p \in [0, 1]$$

The source and target distributions corresponding to $p = k/10$ for $k = 1, 2, 3, 4$ are depicted in Figure 1. For each $k$, the unique optimal transport plan from $\mu_k = \mu(k/10)$ to $\nu$ is given by

$$\Gamma_k = \left\{((-1, -1 + \frac{k}{10}), (-1, 1)), ((1, 1 - \frac{k}{10}), (1, -1))\right\}.$$

For each of these GOT tasks, we draw 200 samples from the source distribution $\mu_k * \mathcal{N}_\sigma$ and target distribution $\nu * \mathcal{N}_\sigma$, and use the empirical $W_2$ distance as an estimate of the true $W_2(\mu_k * \mathcal{N}_\sigma, \nu * \mathcal{N}_\sigma)$. We repeat the process 100 times and report the mean, as shown in the following figures.

By Theorem 4.1 and Theorem 4.4, we expect $W_2^2(\mu_k, \nu) - W_2^2(\mu_k * \mathcal{N}_\sigma, \nu * \mathcal{N}_\sigma)$ to be of scale $e^{-c/\sigma^2}$ for $\sigma \in (0, R(\Gamma_k))$, and $W_2^2(\mu_k, \nu) - W_2^2(\mu_k * \mathcal{N}_\sigma, \nu * \mathcal{N}_\sigma) \gtrsim \sigma$ for $\sigma \geq R(\Gamma_k)$. This transition from exponential to linear is visible in Figure 2.

Using Proposition 3.14, we obtain a lower bound on $R(\Gamma_k)$, which we plot with a vertical dashed line in Figure 2. Exponential decay is visible to the left of the dashed lines, as anticipated. Figure 3 shows this behavior more clearly on a logarithmic scale, where we observe that $\log(-\log(W_2(\mu, \nu) - W_2(\mu * \mathcal{N}_\sigma, \nu * \mathcal{N}_\sigma)))$ is linear in $\log(\sigma)$ for small $\sigma$.

# 7   Conclusion

This paper develops approximation results for Gaussian-smoothed optimal transport, showing that GOT approximates the Wasserstein distance exponentially well for small $\sigma$ when $\mu$ and $\nu$ possess a unique perfect matching. By contrast, if $\mu$ and $\nu$ do not possess a unique perfect matching, then the gap between $W_2(\mu, \nu)$ and $W_2(\mu * \mathcal{N}_\sigma, \nu * \mathcal{N}_\sigma)$ is linear in $\sigma$ as $\sigma \to 0$. The difference between these two behaviors can be traced to the fact that if $\mu$ and $\nu$ possess a unique perfect matching, then

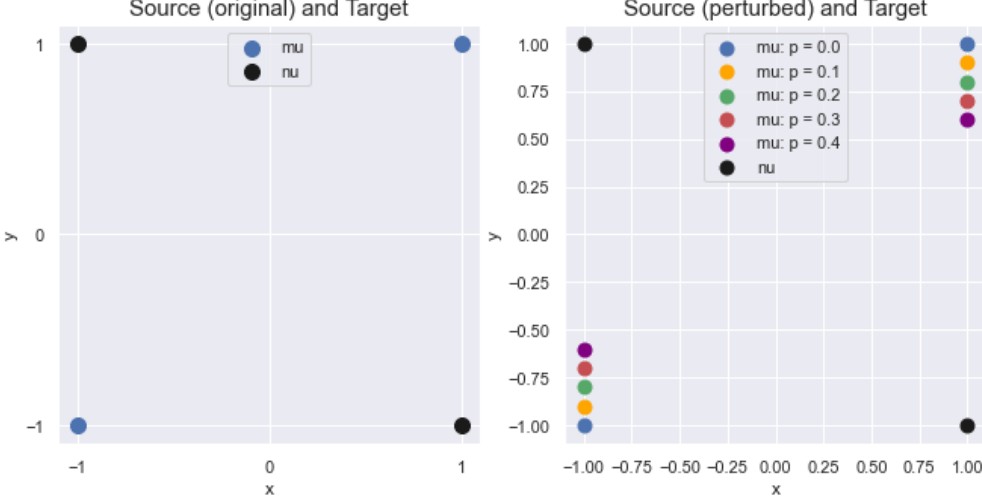

Figure 1: Source and Target distributions

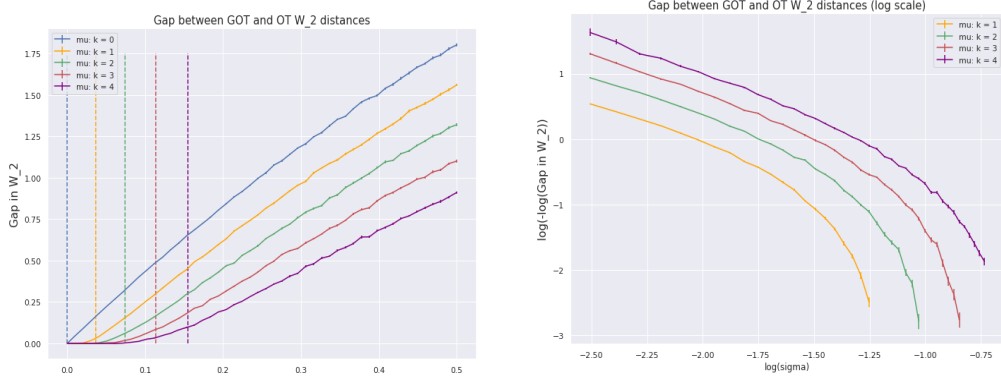

Figure 2: Rate of $W_2(\mu, \nu) - W_2(\mu * \mathcal{N}_\sigma, \nu * \mathcal{N}_\sigma)$ in the vanishing $\sigma$ regime.

Figure 3: Rate of $\log(-\log(G_\sigma))$ versus $\log(\sigma)$ in the vanishing noise regime. Here $G_\sigma :=$ $W_2(\mu, \nu) - W_2(\mu * \mathcal{N}_\sigma, \nu * \mathcal{N}_\sigma)$.

the optimal transport plan between them is stable under small perturbations of $\mu$ and $\nu$. In particular, for noise distributions $Q$ with sufficiently small support, $W_2(\mu, \nu) = W_2(\mu * Q, \nu * Q)$. On the other hand, if $\mu$ and $\nu$ do not possess a unique perfect matching, then their optimal plan is not stable, and can change even under infinitesimal perturbation.

Our techniques are based on a new notion of stability for discrete optimal transport plans, which we call *robustness of optimality* and characterize by developing strong variants of the classic notions of cyclical monotonicity and implementability for optimal transport plans. Just as cyclical monotonicity is closely connected to the convexity of the dual potentials for the optimal transport problem, we show that strong cyclical montonicity is equivalent to *strong* convexity. This characterization shows that a discrete optimal transport plan is robust to perturbations of the source and target measures if and only if its support lies in the subdifferential of a strongly convex function. We anticipate that this characterization will have statistical and computational implications for the estimation of optimal transport plans between discrete measures.

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
