# Asymptotics of smoothed Wasserstein distances in the small noise regime
## Supplementary Material

**Yunzi Ding**[1]    **Jonathan Niles-Weed**[2]
[1]Courant Institute of Mathematical Sciences, NYU
[2]Courant Institute of Mathematical Sciences and the Center for Data Science, NYU
`yunziding@gmail.com`
`jnw@cims.nyu.edu`

## Contents

## 1  Omitted proofs for Section 3

### 1.1  Proof of Proposition 3.6

*Proof.* Suppose $\Gamma$ is $f$-strongly cyclically monotone for some positive residual $f$. Denote

$$M := \max\left\{ \max_i \|x_i\|, \max_i \|y_i\| \right\}.$$

We will show that $\Gamma$ is $\epsilon$-robust for any $\epsilon > 0$ satisfying

$$4M\epsilon < \min_{i \neq j} f(i, j).$$

In fact, for any distinct $\tau(1), \tau(2), \ldots, \tau(n) \in [k]$, by the definition of $f$-strong cyclical monotonicity,

$$\sum_{i=1}^{n} \langle x_{\tau(i)}, y_{\tau(i)} - y_{\tau(i+1)} \rangle \geq \sum_{i=1}^{n} f(\tau(i), \tau(i+1))$$

Thus for any choice of $\alpha_{\tau(1)}, \ldots, \alpha_{\tau(n)}$ such that $\max \|\alpha_{\tau(i)}\| \leq \epsilon$, we have

$$\frac{1}{2} \sum_{i=1}^{n} \|(x_{\tau(i)} + \alpha_{\tau(i)}) - (y_{\tau(i+1)} + \alpha_{\tau(i+1)})\|^2 - \frac{1}{2} \sum_{i=1}^{n} \|x_{\tau(i)} - y_{\tau(i)}\|^2$$

$$= \sum_{i=1}^{n} \langle x_{\tau(i)}, y_{\tau(i)} - y_{\tau(i+1)} \rangle + \sum_{i=1}^{n} \langle \alpha_{\tau(i)}, x_{\tau(i)} - x_{\tau(i-1)} + y_{\tau(i)} - y_{\tau(i+1)} \rangle + \frac{1}{2} \sum_{i=1}^{n} \|\alpha_{\tau(i)} - \alpha_{\tau(i+1)}\|^2$$

$$\geq \sum_{i=1}^{n} f(\tau(i), \tau(i+1)) - 4nM\epsilon$$

$$> 0.$$

Hence $R(\Gamma) > 0$.

On the other hand, given $R(\Gamma) > 0$, we show that $\Gamma$ is the unique optimal transport plan from $\{x_i\}$ to $\{y_i\}$. We prove by contradiction. If $\Gamma$ is not unique, then there exists distinct $\tau(1), \ldots, \tau(n) \in [k]$ such that

$$\sum_{i=1}^{n} \|x_{\tau(i)} - y_{\tau(i)}\|^2 = \sum_{i=1}^{n} \|x_{\tau(i)} - y_{\tau(i+1)}\|^2. \tag{1}$$

Since $R(\Gamma) > 0$, for $\epsilon_0 = R(\Gamma)/2$ and any choice of $\tau(1), \ldots, \tau(n)$ with $\|\tau(i)\| \leq \epsilon_0$, we have

$$\sum_{i=1}^{n} \|x_{\tau(i)} - y_{\tau(i)}\|^2 \leq \sum_{i=1}^{n} \|(x_{\tau(i)} + \alpha_{\tau(i)}) - (y_{\tau(i+1)} + \alpha_{\tau(i+1)})\|^2.$$

Specifically, for any $j \in [n]$, letting $\tau(i) = 0$ for all $i \neq j$ in the above equation gives

$$2\langle \alpha_{\tau(j)}, x_{\tau(j)} - y_{\tau(j+1)} \rangle \leq \|\alpha_{\tau(j)}\|^2$$

for any $\alpha_{\tau(j)} \in \mathbb{R}^d$ with $\|\alpha_{\tau(j)}\| \leq \epsilon_0$. Therefore we must have

$$x_{\tau(j)} = y_{\tau(j+1)}, \quad \forall j \in [k].$$

Using (1), we also know that

$$x_{\tau(j)} = y_{\tau(j)}, \quad \forall j \in [k],$$

which violates the assumption that $\{y_i\}$ are distinct points in $\mathbb{R}^d$. Thus we conclude that $\Gamma$ is unique; hence it is also strongly cyclically monotone due to Proposition 3.8. $\qquad\square$

## 1.2 Proof of Proposition 3.8

*Proof.* (i) to (ii). The idea is borrowed from [1, 2, 3]. Suppose $\Gamma$ is $f$-strongly cyclically monotone for a positive residual function $f$. For $i \in [k]$, denote

$$v_i := \inf_{\substack{\theta(1)=1, \theta(n+1)=i, \\ \theta(2), \ldots, \theta(n) \in [k], \\ \theta(s) \neq \theta(s+1)}} \left( \sum_{s=1}^{n} \langle x_{\theta(s)}, y_{\theta(s)} - y_{\theta(s+1)} \rangle - \sum_{s=1}^{n} f(\theta(s), \theta(s+1)) \right)$$

By the $f$-strong cyclical monotonicity, we have $v_1 \geq 0$. Furthermore, for $i > 1$ and any sequence $\{\theta(s)\}$ with $\theta(1) = 1$, $\theta(n+1) = i$ and $\theta(s) \neq \theta(s+1)$, there holds

$$\sum_{s=1}^{n} \langle x_{\theta(s)}, y_{\theta(s)} - y_{\theta(s+1)} \rangle + \langle x_i, y_i - y_1 \rangle \geq \sum_{s=1}^{n} f(\theta(s), \theta(s+1)) + f(i, 1)$$

and it follows that

$$v_i \geq f(i, 1) - \langle x_i, y_i - y_1 \rangle > -\infty.$$

For any $j \neq i$ and any fixed $\epsilon > 0$, there exists a sequence $\{\theta(s)\}$ with $\theta(1) = 1$, $\theta(n+1) = i$ and $\theta(s) \neq \theta(s+1)$, such that

$$\sum_{s=1}^{n} \langle x_{\theta(s)}, y_{\theta(s)} - y_{\theta(s+1)} \rangle - \sum_{s=1}^{n} f(\theta(s), \theta(s+1)) \leq v_i + \epsilon. \tag{2}$$

Consider the same $\{\theta(s)\}$ with one more term $\theta(n+2) := j$. By definition of $v_j$ we have

$$v_j \leq \sum_{s=1}^{n} \langle x_{\theta(s)}, y_{\theta(s)} - y_{\theta(s+1)} \rangle + \langle x_i, y_i - y_j \rangle - \sum_{s=1}^{n+1} f(\theta(s), \theta(s+1)) \tag{3}$$

Comparing (2) and (3) we get

$$v_j \leq v_i + \langle x_i, y_i - y_j \rangle - f(i,j) + \epsilon \tag{4}$$

We set $\varphi(x_i) = -v_i$. Letting $\epsilon \downarrow 0$ in (4) yields

$$\langle x_i, y_i - y_j \rangle \geq \varphi(x_i) - \varphi(x_j) + f(i,j).$$

Hence $\Gamma$ is $f$-strongly implementable.

(ii) to (iii). We prove by contradiction. Suppose $\Gamma$ is not the unique optimal transport plan; this means either $\Gamma$ is not optimal or there exists a different coupling $\Gamma'$ with the same cost. Either case, there exists a sequence $\{\theta(s)\}_{s=1}^{n}$ such that

$$\sum_{s=1}^{n} \|x_{\theta(s)} - y_{\theta(s)}\|^2 \geq \sum_{s=1}^{n} \|x_{\theta(s)} - y_{\theta(s+1)}\|^2$$

Summing over $s$, we get

$$\sum_{s=1}^{n} f(\theta(s), \theta(s+1)) \leq \sum_{s=1}^{n} \langle x_{\theta(s)}, y_{\theta(s)} - y_{\theta(s+1)} \rangle$$

$$= \frac{1}{2} \left( \sum_{s=1}^{n} \|x_{\theta(s)} - y_{\theta(s+1)}\|^2 - \sum_{s=1}^{n} \|x_{\theta(s)} - y_{\theta(s)}\|^2 \right)$$

$$\leq 0,$$

a contradiction.

(iii) to (i). Suppose $\Gamma$ is the unique optimal transport plan from $\{x_i\}$ to $\{y_i\}$. Denote $c_0$ the transport cost of $\Gamma$. For any other transport plan in the form of a bijection between $\{x_i\}$ and $\{y_i\}$, denote $c_1$ the minimum among their costs, then $c_1 > c_0$. Choose a small enough $\lambda > 0$, such that for any choice of $\tau(1), \tau(2), \ldots, \tau(n) \in [k]$ with no duplicates, there holds

$$\frac{\lambda}{2} \sum_{i=1}^{n} \|y_{\tau(i)} - y_{\tau(i+1)}\|^2 \leq c_1 - c_0.$$

Now for $f(i,j) = \frac{\lambda}{2}\|y_i - y_j\|^2$ we have

$$\sum_{i=1}^{n} \|x_{\tau(i)} - y_{\tau(i+1)}\|^2 - \sum_{i=1}^{n} \|x_{\tau(i)} - y_{\tau(i)}\|^2 \geq c_1 - c_0 \geq \sum_{i=1}^{n} f(\tau(i), \tau(i+1)).$$

If there are duplicates in $(\tau(1), \tau(2), \ldots, \tau(n))$, we break the loop $\tau(1) \to \tau(2) \to \cdots \to \tau(n) \to \tau(1)$ into separate loops without duplicates, apply the above inequality to each loop and sum them up. We conclude by definition that $\Gamma$ is $f$-strongly cyclically monotone. $\square$

## 1.3 Proof of Proposition 3.13

*Proof of Proposition 3.13.* We only need to show that, for an $\epsilon$ satisfying (7), and any choice of $\tau(1), \tau(2), \ldots, \tau(n) \in [k]$, and $\alpha(1), \ldots, \alpha(n)$ with $\|\alpha(i)\| \leq \epsilon$, there holds

$$\sum_{i} \|x_{\tau(i)} - y_{\tau(i)}\|^2 \leq \sum_{i} \|(x_{\tau(i)} + \alpha_{\tau(i)}) - (y_{\tau(i+1)} + \alpha_{\tau(i+1)})\|^2. \tag{5}$$

In fact, (5) is equivalent to

$$2\sum_i \langle \alpha_{\tau(i)}, y_{\tau(i+1)} - y_{\tau(i)} + x_{\tau(i-1)} - x_{\tau(i)} \rangle \leq 2\sum_i \langle x_{\tau(i)}, y_{\tau(i)} - y_{\tau(i+1)} \rangle + \sum_i \|\alpha_{\tau(i)} - \alpha_{\tau(i+1)}\|^2$$

(6)

Since $\|\alpha(i)\| \leq \epsilon$ for all $i$, we have

$$2\sum_i \langle \alpha_{\tau(i)}, y_{\tau(i+1)} - y_{\tau(i)} + x_{\tau(i-1)} - x_{\tau(i)} \rangle$$

$$\leq 2\sum_i \epsilon \cdot \left( \|y_{\tau(i+1)} - y_{\tau(i)}\| + \|x_{\tau(i+1)} - x_{\tau(i)}\| \right)$$

$$\leq \sum_i f(\tau(i), \tau(i+1))$$

where we used the choice of $\epsilon$ in the last inequality. In the meantime, strong implementability gives

$$2\sum_i \langle x_{\tau(i)}, y_{\tau(i)} - y_{\tau(i+1)} \rangle + \sum_i \|\alpha_{\tau(i)} - \alpha_{\tau(i+1)}\|^2 \geq \sum_i f(\tau(i), \tau(i+1)).$$

Therefore (6) holds, which completes the proof. □

### 1.4  Proof of Proposition 3.14

*Proof.* Following the proof of Proposition 3.13, we only need to show that, for the residual $f(i,j)$ defined in Theorem 3.10, there holds

$$2\sum_i \epsilon \cdot \left( \|y_{\tau(i+1)} - y_{\tau(i)}\| + \|x_{\tau(i+1)} - x_{\tau(i)}\| \right) \leq \sum_i f(\tau(i), \tau(i+1)).$$  (7)

By the choice of $\epsilon$, we have

$$2\sum_i \epsilon \cdot \left( \|y_{\tau(i+1)} - y_{\tau(i)}\| + \|x_{\tau(i+1)} - x_{\tau(i)}\| \right)$$

$$\leq \sum_i \max \left\{ \frac{1}{\beta} \|x_{\tau(i+1)} - x_{\tau(i)}\|^2, \alpha \|y_{\tau(i+1)} - y_{\tau(i)}\|^2 \right\}.$$

Meanwhile,

$$\sum_i f(\tau(i), \tau(i+1))$$

$$= \frac{1}{\beta - \alpha} \sum_i \left( \|x_{\tau(i)} - x_{\tau(i+1)}\|^2 + \alpha\beta \|y_{\tau(i)} - y_{\tau(i+1)}\|^2 - 2\alpha \langle y_{\tau(i)} - y_{\tau(i+1)}, x_{\tau(i)} - x_{\tau(i+1)} \rangle \right)$$

$$\geq \frac{1}{\beta - \alpha} \sum_i \left( \|x_{\tau(i)} - x_{\tau(i+1)}\|^2 + \alpha\beta \|y_{\tau(i)} - y_{\tau(i+1)}\|^2 - \alpha \left( \lambda \|x_{\tau(i)} - x_{\tau(i+1)}\|^2 + \frac{1}{\lambda} \|y_{\tau(i)} - y_{\tau(i+1)}\|^2 \right) \right).$$

The last inequality holds for any $\lambda > 0$ by the Cauchy-Schwarz inequality. Choosing $\lambda = 1/\beta$ and $\lambda = 1/\alpha$ yields

$$\sum_i f(\tau(i), \tau(i+1)) \geq \max \left\{ \frac{1}{\beta} \|x_{\tau(i+1)} - x_{\tau(i)}\|^2, \alpha \|y_{\tau(i+1)} - y_{\tau(i)}\|^2 \right\}.$$

Therefore (7) holds, which completes the proof. □

## 2  Omitted proofs for Section 4

### 2.1  Proof of Theorem 4.1

*Proof.* Define the truncated smoothing kernel

$$\tilde{\mathcal{N}}_\sigma := \mathcal{N}(0, \sigma^2 I) \cdot \mathbf{1}\{\|X\| \leq \epsilon_*\} + (1 - p)\delta_0$$

where
$$p = \mathbb{P}\left[\|\mathcal{N}(0, \sigma^2 I)\| < \epsilon_*\right].$$
Since $\tilde{\mathcal{N}}_\sigma$ is supported on $B(0, \epsilon_*)$, by Lemma 4.2, we know
$$W_2(\mu * \tilde{\mathcal{N}}_\sigma, \nu * \tilde{\mathcal{N}}_\sigma) = W_2(\mu, \nu).$$
Therefore,
$$
\begin{aligned}
&|W_2(\mu * \mathcal{N}_\sigma, \nu * \mathcal{N}_\sigma) - W_2(\mu, \nu)|^2 \\
&= |W_2(\mu * \mathcal{N}_\sigma, \nu * \mathcal{N}_\sigma) - W_2(\mu * \tilde{\mathcal{N}}_\sigma, \nu * \tilde{\mathcal{N}}_\sigma)|^2 \\
&\leq (W_2(\mu * \mathcal{N}_\sigma, \mu * \tilde{\mathcal{N}}_\sigma) + W_2(\nu * \mathcal{N}_\sigma, \nu * \tilde{\mathcal{N}}_\sigma))^2 \\
&\lesssim \mathbb{E}_{z \sim \mathcal{N}(0, \sigma^2 I)}\left[\|z\|^2 \mathbf{1}_{\|z\| \geq \sigma_*}\right] \\
&= \sigma^2 \, \mathbb{E}_{z \sim \mathcal{N}(0, I)}\left[\|z\|^2 \mathbf{1}_{\|z\| \geq \sigma_*/\sigma}\right] \\
&\lesssim \sigma \sigma_* e^{-\sigma_*^2/2\sigma^2}.
\end{aligned}
$$

Here the second inequality is yielded by considering a coupling of $\mu * \mathcal{N}_\sigma$ and $\mu * \tilde{\mathcal{N}}_\sigma$ that is the distribution of $(X + Z, X + Z \cdot \mathbf{1}\{\|Z\| \leq \epsilon_*\})$, where $X$ and $Z$ are independent, $X \sim \mu$ and $Z \sim \mathcal{N}(0, \sigma^2 I)$, and the same coupling for $\mu$ replaced with $\nu$. Taking square root on both sides yields the result. $\qquad\square$

## 2.2 Proof of Lemma 4.2

*Proof.* We naturally split the source measure into $k$ parts:
$$\mu * Q = \sum_{i=1}^{k}\left(\frac{1}{k}\delta(x_i) * Q\right)$$
Consider a map $T$ which, for each $i \in [k]$, is defined by
$$T(x) = x + y_i - x_i \qquad \forall x \in B(x_i, \sigma_*).$$
We can obtain a transport plan between $\mu * Q$ and $\nu * Q$ by considering the distribution of a pair of random variables $(X, T(X))$ for $X \sim \mu * Q$. The support of this plan lies in the set $\bigcup_{i=1}^{k} \bigcup_{\alpha \in B(0, \sigma_*)}(x_i + \alpha, y_i + \alpha)$. By the definition of $R(\Gamma)$, this set is cyclically monotone, so this coupling is optimal for $\mu * Q$ and $\nu * Q$ by Theorem 3.2. Therefore
$$
\begin{aligned}
W_2^2(\mu * Q, \nu * Q) &= \int \|x - T(x)\|^2 d(\mu * Q)(x) \\
&= \frac{1}{k}\sum_{i=1}^{k} \|y_i - x_i\|^2 = W_2^2(\mu, \nu),
\end{aligned}
$$
as claimed. $\qquad\square$

## 2.3 Proof of Proposition 4.3

*Proof.* For $M > 0$, denote
$$g(m) := \sup\left\{\sum_{i=1}^{n} \|x_{\tau(i)} - y_{\tau(i)}\|^2 - \sum_{i=1}^{n} \|(x_{\tau(i)} + \alpha_{\tau(i)}) - (y_{\tau(i+1)} + \alpha_{\tau(i+1)})\|^2 : \max_i \|\alpha_{\tau(i)}\| = m\right\},$$
then $G(M) = \sup\{g(m) : m \in [0, M]\}$. We first prove that $g(m)$ is concave in $m$. In fact, denote the set
$$\mathcal{I} = \left\{(\tau(1), \ldots, \tau(n), \alpha_{\tau(1)}, \ldots, \alpha_{\tau(n)}) : \tau(i) \in [k], \ \tau(i) \neq \tau(j), \ \max_i \|\alpha_{\tau(i)}\| = 1\right\}.$$
By definition,
$$
\begin{aligned}
g(m) = \sup\Big\{&\sum_{i=1}^{n} \|x_{\tau(i)} - y_{\tau(i)}\|^2 - \sum_{i=1}^{n} \|(x_{\tau(i)} + m\alpha_{\tau(i)}) - (y_{\tau(i+1)} + m\alpha_{\tau(i+1)})\|^2 : \\
&(\tau(1), \ldots, \tau(n), \alpha_{\tau(1)}, \ldots, \alpha_{\tau(n)}) \in \mathcal{I}\Big\}
\end{aligned}
$$

Note that, for every choice of $(\tau(1), \ldots, \tau(n))$ and $\alpha_{\tau(1)}, \ldots, \alpha_{\tau(n)}) \in \mathcal{I}$,

$$\sum_{i=1}^{n} \|x_{\tau(i)} - y_{\tau(i)}\|^2 - \sum_{i=1}^{n} \|(x_{\tau(i)} + m\alpha_{\tau(i)}) - (y_{\tau(i+1)} + m\alpha_{\tau(i+1)})\|^2$$

is a concave function in $m$. Therefore, $g(m)$ is concave in $m$, and $G(M)$ is also concave in $M$. $\quad\square$

## 2.4 Proof of Theorem 4.4

*Proof.* For $M > \sigma_*$, pick $\tau(1), \tau(2), \ldots, \tau(n) \in [k]$ and $\{\alpha_{\tau(i)}\}_{i=1}^{n} \subset \mathbb{R}^d$ such that $\|\alpha_{\tau(i)}\| \leq M$ and

$$\sum_{i=1}^{n} \|x_{\tau(i)} - y_{\tau(i)}\|^2 - \sum_{i=1}^{n} \|(x_{\tau(i)} + \alpha_{\tau(i)}) - (y_{\tau(i+1)} + \alpha_{\tau(i+1)})\|^2 = G(M).$$

For every $i \in [k]$, denote $B_{\tau(i)}$ the ball centered at $x_{\tau(i)} + \alpha_{\tau(i)}$ with radius $\sigma$, and $\hat{B}_{\tau(i)}$ the ball centered at $y_{\tau(i)} + \alpha_{\tau(i)}$ with radius $\sigma$. Also denote

- $\gamma \in \Pi(\mu * \mathcal{N}_\sigma, \nu * \mathcal{N}_\sigma)$ the law of $(X + Z, Y + Z)$, where $(X, Y) \sim \frac{1}{k} \sum_{i=1}^{k} \delta(x_i, y_i)$ and $Z \sim \mathcal{N}_\sigma$ are independent.

- $\gamma_{\tau(i)} \in \Pi(\mathsf{Unif}(B_{\tau(i)}), \mathsf{Unif}(\hat{B}_{\tau(i)}))$ the coupling associated with the transport map

$$x \mapsto x + y_{\tau(i)} - x_{\tau(i)};$$

- $\tilde{\gamma}_{\tau(i)} \in \Pi(\mathsf{Unif}(B_{\tau(i)}), \mathsf{Unif}(\hat{B}_{\tau(i+1)}))$ the coupling associated with the transport map

$$x \mapsto x + y_{\tau(i+1)} - x_{\tau(i)};$$

- A constant $m = c_d \exp\left(-\frac{(M+\sigma)^2}{2\sigma^2}\right)$, where $c_d$ is a constant only dependent on the dimension $d$.

Consider the following measure in $\mathbb{R}^d \times \mathbb{R}^d$:

$$\tilde{\gamma} := \gamma - m \sum_{i=1}^{n} \gamma_{\tau(i)} + m \sum_{i=1}^{n} \tilde{\gamma}_{\tau(i)}.$$

We shall show that $\tilde{\gamma} \in \Pi(\mu * \mathcal{N}_\sigma, \nu * \mathcal{N}_\sigma)$. We first verify that $\tilde{\gamma}$ is a positive measure on $\mathbb{R}^d \times \mathbb{R}^d$. In fact, for $x, y \in \mathbb{R}^d$,

$$\gamma(dx, dy) = \frac{1}{k} \sum_{i=1}^{k} \left( \frac{1}{(\sqrt{2\pi}\sigma)^d} e^{-\frac{\|x-x_i\|^2}{2\sigma^2}} dx \cdot \delta_{x-x_i+y_i}(dy) \right).$$

Meanwhile,

$$\left( m \sum_{i=1}^{n} \gamma_{\tau(i)} \right) (dx, dy) = m \sum_{i=1}^{n} \left( \frac{\mathbf{1}\{x \in B_{\tau(i)}\}}{\mathsf{Vol}(B_{\tau(i)})} dx \cdot \delta_{x-x_{\tau(i)}+y_{\tau(i)}}(dy) \right).$$

For every $\tau(i)$ such that $x \in B_{\tau(i)}$, note that

$$\|x - x_{\tau(i)}\| \leq \|x - (x_{\tau(i)} + \alpha_{\tau(i)})\| + \|\alpha_{\tau(i)}\| \leq \sigma + M,$$

hence (with a proper choice of $c_d$)

$$\frac{1}{k} \frac{1}{(\sqrt{2\pi}\sigma)^d} e^{-\frac{\|x-x_{\tau(i)}\|^2}{2\sigma^2}} \geq \frac{1}{k} \frac{1}{(\sqrt{2\pi}\sigma)^d} e^{-\frac{(M+\sigma)^2}{2\sigma^2}} \geq \frac{m}{\mathsf{Vol}(B_{\tau(i)})}.$$

As a result, $\gamma - m \sum_{i=1}^{n} \gamma_{\tau(i)} \geq 0$, and $\tilde{\gamma}$ is a positive measure. Also note that its first marginal (i.e. the marginal on the first $d$ dimensions) and second marginal (i.e. the marginal on the last $d$

dimensions) agree with the respective marginals of $\gamma$. Thus we conclude that $\tilde{\gamma} \in \Pi(\mu * \mathcal{N}_\sigma, \nu * \mathcal{N}_\sigma)$. Now note that

$$\int c(x,y) d\gamma(x,y) - \int c(x,y) d\tilde{\gamma}(x,y)$$

$$= m \left( \sum_{i=1}^{n} \|x_{\tau(i)} - y_{\tau(i)}\|^2 - \sum_{i=1}^{n} \|(x_{\tau(i)} + \alpha_{\tau(i)}) - (y_{\tau(i+1)} + \alpha_{\tau(i+1)})\|^2 \right)$$

$$= m \cdot G(M).$$

In the meantime,

$$\int c(x,y) d\gamma(x,y) = \frac{1}{2k} \sum_{i=1}^{k} \|x_i - y_i\|^2 = W_2^2(\mu, \nu),$$

therefore,

$$W_2^2(\mu * \mathcal{N}_\sigma, \nu * \mathcal{N}_\sigma)$$

$$\leq \int c(x,y) d\tilde{\gamma}(x,y)$$

$$\leq W_2^2(\mu, \nu) - G(M) \cdot c_d \exp\left( -\frac{(M+\sigma)^2}{2\sigma^2} \right).$$

In particular, choosing $M = \sigma + \sigma_*$ yields

$$W_2^2(\mu, \nu) - W_2^2(\mu * \mathcal{N}_\sigma, \nu * \mathcal{N}_\sigma) \gtrsim G(\sigma + \sigma_*) \exp\left( -c \frac{\sigma_*^2}{\sigma^2} \right).$$

The rest follows from the observation that, for $\sigma \in (0, 2\sigma_*)$,

$$G(\sigma + \sigma_*) = G(\sigma + \sigma_*) - G(\sigma_*) \geq \frac{G(3\sigma_*) - G(\sigma_*)}{2\sigma_*} \cdot \sigma$$

since $G$ is concave by Proposition 4.3. $\qquad \square$

## 3 Omitted proofs for Section 5

### 3.1 Proof of Theorem 5.1

*Proof.* Suppose that there exists a transport plan $\pi$ between $\mu$ and $\nu$ which achieves the optimal cost and is not a perfect matching. Without loss of generality, we assume that $(x_1, y_1)$ and $(x_1, y_2)$ both lie in the support of $\pi$. Let $\lambda = \min\{\pi(x_1, y_1), \pi(x_1, y_2)\}$. We decompose $\mu$ and $\nu$ as

$$\hat{\mu} = \mu - 2\lambda\delta(x_1), \quad \tilde{\mu} = 2\lambda\delta(x_1),$$
$$\hat{\nu} = \nu - \lambda\left(\delta(y_1) + \delta(y_2)\right), \quad \tilde{\nu} = \lambda\left(\delta(y_1) + \delta(y_2)\right).$$

By Lemma 5.2, there exists $c_0 > 0$ such that for $\sigma \in (0, c_0)$,

$$W_2^2(\tilde{\mu}, \tilde{\nu}) - W_2^2(\tilde{\mu} * \mathcal{N}_\sigma, \tilde{\nu} * \mathcal{N}_\sigma) \gtrsim \sigma.$$

Therefore, for $\sigma \in (0, c_0)$, we also have

$$W_2^2(\mu, \nu) - W_2^2(\mu * \mathcal{N}_\sigma, \nu * \mathcal{N}_\sigma)$$
$$\geq W_2^2(\hat{\mu}, \hat{\nu}) - W_2^2(\hat{\mu} * \mathcal{N}_\sigma, \hat{\nu} * \mathcal{N}_\sigma) + W_2^2(\tilde{\mu}, \tilde{\nu}) - W_2^2(\tilde{\mu} * \mathcal{N}_\sigma, \tilde{\nu} * \mathcal{N}_\sigma)$$
$$\geq W_2^2(\tilde{\mu}, \tilde{\nu}) - W_2^2(\tilde{\mu} * \mathcal{N}_\sigma, \tilde{\nu} * \mathcal{N}_\sigma)$$
$$\gtrsim \sigma.$$

$\qquad \square$

### 3.2 Proof of Lemma 5.2

*Proof.* First suppose that $x, y_1, y_2$ are not on the same line with $y_1$ between $x$ and $y_2$ or $y_2$ between $x$ and $y_1$. Let $\Delta$ be the bisecting hyperplane of $\angle y_1 x y_2$, namely

$$\Delta = \left\{ z \in \mathbb{R}^d \ : \ \frac{\langle z - x, y_1 - x \rangle}{|y_1 - x|} = \frac{\langle z - x, y_2 - x \rangle}{|y_2 - x|} \right\},$$

and define its unit normal vector $\mathbf{m}$ such that $\langle \mathbf{m}, y_1 - x \rangle > 0$. We adopt the decomposition

$$\begin{aligned}
\mu_+ &:= \mathcal{N}(x, \sigma^2) \mid \langle z - x, \mathbf{m} \rangle > 0, \\
\mu_- &:= \mathcal{N}(x, \sigma^2) \mid \langle z - x, \mathbf{m} \rangle < 0,
\end{aligned} \tag{8}$$

and

$$\begin{aligned}
\nu_{1+} &:= \mathcal{N}(y_1, \sigma^2) \mid \langle z - y_1, \mathbf{m} \rangle > 0, \\
\nu_{1-} &:= \mathcal{N}(y_1, \sigma^2) \mid \langle z - y_1, \mathbf{m} \rangle < 0, \\
\nu_{2+} &:= \mathcal{N}(y_2, \sigma^2) \mid \langle z - y_2, \mathbf{m} \rangle > 0, \\
\nu_{2-} &:= \mathcal{N}(y_2, \sigma^2) \mid \langle z - y_2, \mathbf{m} \rangle < 0.
\end{aligned} \tag{9}$$

Note that all the six sub-probability measures above have mass $1/2$. By the definition of $W_2$, we have

$$W_2^2(\mu_0 * \mathcal{N}_\sigma, \nu_0 * \mathcal{N}_\sigma) \le \frac{1}{2} \left( W_2^2(\mu_+, \nu_{1+}) + W_2^2(\mu_+, \nu_{1-}) + W_2^2(\mu_-, \nu_{2+}) + W_2^2(\mu_-, \nu_{2-}) \right). \tag{10}$$

It is obvious that

$$W_2^2(\mu_+, \nu_{1+}) = \frac{1}{2} \| x - y_1 \|^2, \quad W_2^2(\mu_-, \nu_{2-}) = \frac{1}{2} \| x - y_2 \|^2.$$

For $W_2^2(\mu_+, \nu_{1-})$, consider the map

$$T_\#(x + t) \ = \ y_1 - t, \quad t \sim \mathcal{N}(0, \sigma^2 I)$$

we have

$$\begin{aligned}
W_2^2(\mu_+, \nu_{1-}) &\le \mathbb{E}_{u \sim \mu_+} \| u - T_\# u \|^2 \\
&= \mathbb{E}_{u \sim \mu_+} \| u - (y_1 - u + x) \|^2 \\
&= \frac{1}{2} \| x - y_1 \|^2 - 4 \mathbb{E}_{u \sim \mu_+} \langle y_1 - x, u - x \rangle + 4 \mathbb{E}_{u \sim \mu_+} \| u - x \|^2 \\
&= \frac{1}{2} \| x - y_1 \|^2 - 4 c_1 \sigma \langle \mathbf{m}, y_1 - x \rangle + 4 c_2 \sigma^2,
\end{aligned}$$

where $c_1$ and $c_2$ are absolute positive constants. Similarly,

$$W_2^2(\mu_-, \nu_{2+}) \le \frac{1}{2} \| x - y_2 \|^2 - 4 c_1 \sigma \langle \mathbf{m}, x - y_2 \rangle + 4 c_2 \sigma^2.$$

Plugging into (10) we get

$$W_2^2(\mu_0 * \mathcal{N}_\sigma, \nu_0 * \mathcal{N}_\sigma) \le W_2^2(\mu_0, \nu_0) - 4 c_1 \sigma \langle \mathbf{m}, y_1 - y_2 \rangle + 8 c_2 \sigma^2,$$

hence $W_2^2(\mu_0, \nu_0) - W_2^2(\mu_0 * \mathcal{N}_\sigma, \nu_0 * \mathcal{N}_\sigma) \gtrsim \sigma$ for small $\sigma$, since $\langle \mathbf{m}, y_1 - y_2 \rangle > 0$.

Finally, we consider the special case where $x, y_1, y_2$ are on the same line and $y_1$ is between $x$ and $y_2$. We choose $\mathbf{m}$ the unit vector along the direction $x - y_1$, and the same line of proof yields the conclusion. $\square$