# OpenReview forum: "Asymptotics of smoothed Wasserstein distances in the small noise regime"
_NeurIPS.cc/2022/Conference — NeurIPS 2022 Accept_

### Official Review · Reviewer_jBuP · 2022-07-06

**Rating:** 7
**Confidence:** 3
**Soundness:** 3 good
**Presentation:** 2 fair
**Contribution:** 3 good

**Summary:**

This paper presents an analysis of the Gaussian-smoothed Wasserstein distance (GOT) in the framework where the Gaussian kernel parameter $\sigma$ is "small". It is already known that (GOT) approximates the true Wasserstein distance and that the difference between the two is of order $\sigma$. The objective of this paper is to refine this bound and to show, under certain assumptions of uniqueness of the transport plan, that this bound can be improved and that GOT approximates Wasserstein exponentially well in a certain regime. More precisely, the authors show that there is a phase transition on $\sigma$ such that, below, the bound is exponential, and, above, it is linear in $\sigma$. This paper completes the understanding of GOT with respect to $\sigma$, which has already been studied in the $\sigma \to +\infty$ regime.

**Questions:**

- In Remark 2.14 it is said that condition (5) in the context of residual functions of quadratic type is "easier to verify in pratice". It is not really clear to me how (6) can be verified easily because it depends on $\alpha,\beta$. To check that this condition does not hold we should do it for all $\alpha <\beta$ am I correct ? Can the authors detail this point ?
- The fact that $\Gamma$ is f-strongly implementable seems to be really related to the article [PdC20] of the paper, i.e. to the fact of imposing that the Monge map is the gradient of a smooth and strongly convex potential. Can the authors discuss this connection in more detail?
- The proof of Theorem 2.7 is rather succinct: "This is a direct application of Theorem 4 in [THG17]". Could the authors give more details about the connection between "$\Gamma$ is strongly implementable" and the results of [THG17] ?

**Limitations:**

The authors did not discuss the potential negative societal impacts of their work, however this is not really relevant in this context as the article is quite theoretical and specific. Concerning the limitations: I find that the authors could discuss how, in practice, can we check if we are in the regime of fast approximation (i.e. when the transport plan is unique).


------- AFTER REBUTTAL -------

As written below I am satisfied with the authors' answer so that I change my score to 7

**Strengths And Weaknesses:**

Overall I find this article quite well written, the thread of definitions and proofs is clear, the ideas are well linked. From a purely theoretical point of view the results are, I think, really interesting. They complete the statistical understanding of GOT with respect to $\sigma$, which is a nice contribution. Moreover, the ideas/tricks introduced and the theorems go far beyond the study of the GOT distance and can certainly be used to establish other theoretical results in optimal transport. In particular I think that the notions of strong implementability/strongly cyclically monotonic and robustness of the transport plans are useful and rich.They allow us to establish the uniqueness of the transport plan, which is, in the discrete case, a key property that is not much addressed by the community as far as I know. The fact of having cleared all these properties around the strong implementability is for me a contribution that is useful in itself. The phase transition of GOT opens also the door to other studies, notably on sample complexity or on approximations of the Wasserstein distance.

The main criticism I would have is that this article focuses on a really specific and technical problem, related to a sub-problem of optimal transport. For a reader who is interested in GOT the contributions are certainly really interesting, but the article does not discuss the potential applications for optimal transport in general neither for machine learning. To be more precise, the article strings together theoretical results without giving much insight, nor discussing the propositions and theorems. It also lacks a conclusion that could perhaps bring some perspectives around this work. In this context, I think it would be interesting to save space by moving the proof of Theorem 3.3 to an appendix to make a conclusion/perspective part and for discussing the different results and their implications. For example it could be interesting to discuss, even informally, the possible generalization to the case of continuous measures or to explain their implications for the Wasserstein distance approximation.

Moreover, the numerical results are very succinct and, I find, difficult to read. I find for example that the phase transition, which is the center of the contributions, is not really visible on Figure 2. I think that this part should be more complete, by better illustrating this phase transition or for example the notion of robustness of a transport plan.

For these reasons I rather recommend a weak-accept, but I am ready to change my mind depending on the authors' answer.

Small remark:
- In terms of notation the $\sigma$ is both used to define a permutation and for the $\sigma$ of the Gaussian kernel.
- Moreover the fact that $\sigma$ is a permutation in Definition 2.4 is not clearly stated in the article.
- Typos: $T$ instead of $\Gamma$ in Proposition 2.12 and 2.13.

---

> ### Author Response · Authors · 2022-08-01
> **Response to reviewer jBuP**
>
> We thank the reviewer for their careful reading of our paper and summarizing of our contributions.
> In the revised version of this paper, we have followed the reviewer's suggestion and added more explanatory text as well as a section with some concluding remarks discussing connections between this concept and similar assumptions that have appeared in the continuous context.
> We believe that these additional remarks give additional justification for focusing on this particular problem: though this problem is rather ``specific,'' as the reviewer notes, we pursued it partially because it seemed to give rise to interesting concepts.
> We agree that the interest of these concepts was left unexplained in our original draft; we hope that our revision has helped to clarify the wider interest of our techniques.
>
> We admit that our numerical experiment in Section 5 is rather limited, and Figure 2 is hard to read due to the small size of the plot. We have included an additional zoomed in plot with data reported on a log scale, which we hope has improved the presentation.
>
> ### Questions
> The reviewer asked why we claimed in Remark 2.14 that (6) is easier than (5) to verify in practice. We agree that this was vague. By ``easier", what we really mean is that the RHS of (6) is neater and requires less calculation than (5) (though they both need $\Theta(k^2)$ time). Here we implicitly assumed knowledge on $\alpha$ and $\beta$, which are inherent to the optimal transport plan $\Gamma$.  We have clarified this in the revision.
>
> The reviewer asked about the connection between our notion of strong implementability and the work [PdC20]. We agree that $f$-strong implementability is closely related to the fact that the Monge map is the gradient of a smooth and strongly convex potential. Our main arguments on this topic is Theorem 2.7 and Corollary 2.9, where we established an equivalence between strong implementability and the existence of a smooth and strongly convex potential, and proposed a representation of the convexity and smoothness parameters of the potential given $f$ of a certain shape.
>
> The reviewer asked for more details on the application of results in [THG17] in the proof of Theorem 2.7. We have expanded this in our revised version. In brief, the condition (i) in Theorem 2.7 is equivalent to the set $\{(y_i, x_i, \varphi(y_i))\}$ being $\mathcal{F}_{\alpha, \beta}$-interpolable in the language of [THG17], and the condition (ii) (equation (3)) in Theorem 2.7 is equivalent to equation (4) in Theorem 4 of [THG17].
>
> ### Remarks
> The reviewer pointed out that the notations for $\sigma$ as both the Gaussian variance parameter and a map from $[n]$ to $[k]$ (for example, in Definition 2.4) can be confusing. We agree and have modified the notation, as well as clarified the requirements on the map. (In fact, it can be limited to cycles.)
>
> We thank the reviewer for finding the typos in Proposition 2.12 and 2.13, where $T$ should be $\Gamma$.
>
> ### Limitations
> The reviewer raised the question that how we can check the uniqueness of an optimal transport plan in practice. We proposed two sufficient conditions in Proposition 2.12 and Proposition 2.13, though we are aware that they both require extra knowledge on the plan $\Gamma$. At the reviewer's suggestion, we have added a remark to the revision indicating a stronger sufficient condition, which is implicit in Proposition 2.6: that the directed bipartite graph with vertex set $\{x_1, \dots, x_k, y_1, \dots, y_k\}$ and arcs between $x_i$ and $y_j$ with weight $\|x_i - y_j\|^2$ should not possess an alternating cycle of zero total cost.

---

> > ### Comment · Reviewer_jBuP · 2022-08-03
> > **Response to authors**
> >
> > I thank the authors for their answer and for their work in the revised manuscript.  I believe now that this is more clear and I like the fact that authors take some time to add a conclusion and some perspectives to their works.  Therefore I will change my score to 7.
> >
> > To conclude a little remark on the form of Figure 1 and 2: if the manuscript is accepted I think it would be good to make Figure 1 more readable by saving it and including it as a pdf and not png/jpg (in the same way as Figure 2) as it is a bit blurred here. Also the titles and xaxis/yaxis of Figure 2 should be made larger. The log xaxis is great we see more clearly what is happening, but the name of yaxis is log(-log(Gap in W2)): is it a typo ? shouldn't it be just log(Gap in W2) ?

---

> > > ### Author Response · Authors · 2022-08-05
> > > **Thank you for your comments**
> > >
> > > We thank the reviewer for the comments. We will remake Figure 1 and enlarge the axes of Figure 2 in our future revision.
> > >
> > > On the $\log-\log$ plot, we meant to plot the $y$-axis as $\log(-\log(W_2(\mu, \nu) - W_2(\mu\ast\mathcal{N}_\sigma, \nu\ast\mathcal{N}_\sigma))$. Theorem 4.1 tells that, in the case of a perfect matching, $W_2(\mu, \nu) - W_2(\mu\ast\mathcal{N}_\sigma, \nu\ast\mathcal{N}_\sigma) \lesssim e^{-c/\sigma^2}$. Therefore, upon taking $\log(-\log(\cdot))$, the gap in $W_2$ would be bounded by a linear function in $\log(\sigma)$, which is evident in the current Figure 3.

---

### Official Review · Reviewer_AtL7 · 2022-07-07

**Rating:** 7
**Confidence:** 4
**Soundness:** 4 excellent
**Presentation:** 3 good
**Contribution:** 3 good

**Summary:**

This paper studies the approximation of the 2-Wasserstein distance between two discrete probability measures $\mu$ and $\nu$ by the 2-Wasserstein distance between the same distributions smoothed by a Gaussian measure, that is called Gaussian-smoothed optimal transport. In particular, they prove the existence of a phase transition in the small noise regime of the variance parameter, which depends on the existence of a perfect matching between the distributions $\mu$ and $\nu$.


**Questions:**

- Could you transpose your results to the case of entropy regularized optimal transport? In this case, the uniqueness of the transport plan is guaranteed but the perfect matching does not make sense anymore for large regularization parameter since the mass of the regularized transport plan is spread out.

- In Proposition 1.1, why is the condition that probability measures are compactly supported essential?


**Limitations:**

The authors decided to limit their study to finitely supported measures, which is well justified.


**Strengths And Weaknesses:**

Studying the behaviour of Gaussian-smoothed OT distance is particularly interest as it approximates the true transport and does not present a curse of dimensionality in the sampling complexity. In particular proving the curious result of the existence of a phase transition in the context of finitely supported measures $\mu$ and $\nu$ is very nice. Moreover, this paper is well written, the problem and presentation are clear and the choices made for the study (the finitely supported framework) are precisely justified. The proofs, based on strong cyclical monotonicity and perfect matching, are quite elegant. The simulation study is also concise and convincing.

Overall, the results constitute a modest (because limited to Gaussian-smoothed OT distance) but very interesting contribution to the study of approximations of the classical Wasserstein distance.

Minor comments:
- The notations for the parameter $\sigma$ of the Gaussian and the permutation $\sigma$ can be confusing.
- The order and presentation of proofs in the appendix could be improved.

---

> ### Author Response · Authors · 2022-08-01
> **Response to reviewer  AtL7**
>
> We thank the reviewer for their careful reading and summarizing of our paper and the supplementary proofs.
>
> ### Minor comments
>  The reviewer pointed out that the notations for $\sigma$ as both the Gaussian variance parameter and the mapping from $[k]$ to $[n]$ can be confusing. We agree and have modified the notation.
>
> The reviewer also commented that the order and presentation of proofs in the appendix could be improved. We thank the reviewer for the suggestion. We have added an organizational summary at the beginning of the appendix to make it easier for the reader to access the proofs.
>
> ### Questions
> The reviewer asked whether our results are transferrable to the case of entropy-regularized optimal transport. We have not considered such extension, but it is a potentially interesting topic for future work. To our knowledge, the Gaussian smoothing framework has largely been considered in the context of the standard (i.e., not entropy-regularized) Wasserstein distance, which motivates our focus in this work.
>
>
> The reviewer also asked why the condition that $\mu$ is compactly supported is essential in Proposition 1.1. Despite the elementary nature of the statement of Proposition 1.1, the proof requires somewhat specialized techniques; in [GGNWP20], the result is derived as a consequence of the fact that $\mu * N_\sigma$ enjoys a log-Sobolev inequality for any compactly supported $\mu$ and any $\sigma > 0$. However, this property can fail to hold when $\mu$ no longer has compact support. Indeed the recent preprint "Rate of convergence of the smoothed empirical Wasserstein distance" (Block et al., arXiv: 2205.02128) shows that there exist non-compactly supported $\mu$ with subgaussian tails for which $W_2(\mu_n * N_\sigma, \mu * N_\sigma) \gg n^{-1/2}$.

---

> > ### Comment · Reviewer_AtL7 · 2022-08-08
> > **Rebuttal**
> >
> > Thanks for the interesting clarification on compactly supported measures, and for taking into account the suggestions in the revised manuscript.

---

### Official Review · Reviewer_Wajc · 2022-07-07

**Rating:** 7
**Confidence:** 4
**Soundness:** 3 good
**Presentation:** 3 good
**Contribution:** 3 good

**Summary:**

The paper provides an approximation rate for Gaussian-smoothed Wasserstein distances for discrete measures. It shows that the approximation rate can be exponential in the perfect-matching plan case (with a phase transition to linear rate if $\sigma$ is large) and is linear otherwise.

**Questions:**

1. The results in the paper are interesting in that the approximation rates for the "perfect-matching" case are much faster than other cases (exponential compared to linear). To complete the story, can you present some sufficient conditions to have perfect matching in OT?

2. The technique used in the paper seems not to be restrictive to discrete measures only. I suggest the author extend these results to arbitrary $\mu$ and $\nu$, as the "perfect-matching" is easier to attain for absolutely continuous probability measure. If it is not possible to do this extension, then it would be helpful to have a paragraph to discuss why.

3. In the statement of Theorem 3.3., what do you mean by $G(\sigma^*)$ is asymptotically greater than $\sigma^*$? Because $\sigma^*$ here is fixed, I can not see the asymptotic greater sign here with respect to what variables.

**Ethics Review Area:**

["I don’t know"]

**Limitations:**

Because of the restriction of the setting in the paper, it is not obvious to see implications in using Gaussian-smoothed Optimal Transport for general measures. I recommend considering the problem in a more general setting and trying to see if the current approach still works.

**Strengths And Weaknesses:**

**Strength**: The results in this paper are interesting and all the theoretical results are technically correct.

**Weakness**: The setting considered in the paper appears to be restrictive. Both measures of interest are assumed to have finite number of support points. It makes the developed theory difficult to apply to general setting of Gaussian-smoothed Optimal Transport.

Some points in presentation that can be improved:

1. In Proposition 1.1., the constant $c$ is dependent on $d$ (dimension) as well. The author should make this point clear so it is coherent with the argument about the curse of dimensionality of the Wasserstein distances.

2. In the presentation of Section 3 (Case I): We need reasons why you choose to present Proof of Theorem 3.3. but not of other results (like Theorem 3.1, which appears to be one of the main results). If the proof of Theorem 3.1. is not presented in the main text, then why do you present Lemma 3.4.? In general, the writing style in Sections 3 and 4 needs to be improved so that the reader can know what proof is going to be presented and what is deferred to the appendix.

3. The order of results in the appendix is not consistent. Dividing those results into several small sections may help to keep track of this issue.

---

> ### Author Response · Authors · 2022-08-01
> **Response to reviewer Wajc**
>
> We thank the reviewer for their careful reading and summarizing of our paper.
>
> We agree with the reviewer that extending our results to measures that do not have finite support is a very interesting question. As we mention in the introduction, the continuous case is much more complicated, with many rates possible between $\sigma$ and $e^{-c/\sigma^2}$, even when the unique optimal coupling is bijective (the continuous analogue of a ``perfect matching''). An example is $\mu = N(0, 1)$ and $\nu = N(0, 4)$, where one can verify that $|W_2(\mu\ast N_\sigma, \nu\ast N_\sigma) - W_2(\mu, \nu)| \asymp \sigma^2$ for small $\sigma$.
> On the other hand, for $\mu = N(0, 1)$ and $\nu = N(c, 1)$ for arbitrary $c \in R$, the error is $0$ for all $\sigma \geq 0$.
> This and other examples suggest that the general (non-atomic) case lacks a clean characterization such as the one we have developed for finite measures.
> We are therefore motivated to focus on the finite support case, which is still of practical interest due to its connection to finite Gaussian mixture models.
>
>
>
> ### On the points in presentation that can be improved.
> * In Proposition 1.1, the constant $c$ is dependent on $d$. We thank the reviewer for pointing this out.
> * In Section 3, we chose to present the proof of Theorem 3.3 in the main text, and deferred the rest to the supplementary material. The main reason is that we wanted to present the proof of both of the main theorems (3.1 and 3.3), but doing so would render the main text over the page limit. Therefore, we included the full proof of Theorem 3.3, and also presented Lemma 3.4 because it is the central idea behind the proof of Theorem 3.1. We thank the reviewer for the suggestion. We have reorganized Sections 3 and 4 in the revision and have deferred all proofs to the appendix.
> * In the appendix, we chose to present the proofs in the same order as the results are presented in the main paper. We have reorganized the appendix and added a table of contents and the beginning to make it easier for the reader to access the proofs.
>
> ### Questions
> * The reviewer asked whether we have sufficient conditions for a perfect matching in an OT problem. We proposed two sufficient conditions in Proposition 2.12 and Proposition 2.13, though we are aware that they both require extra knowledge on the plan $\Gamma$. At the reviewer's suggestion, we have added a remark to the revision indicating a stronger sufficient condition, which is implicit in Proposition 2.6: that the directed bipartite graph with vertex set $\{x_1, \dots, x_k, y_1, \dots, y_k\}$ and arcs between $x_i$ and $y_j$ with weight $\|x_i - y_j\|^2$ should not possess an alternating cycle of zero total cost.
>
> * The reviewer suggested that we extend the results to arbitrary $\mu$ and $\nu$. Please see our response above---even when there exists a ``perfect matching'' in the continuous case, the error can have different polynomial rates, suggesting that a direct extension of our techniques to the non-atomic case does not succeed.
> * The reviewer asked about the meaning of $G(3\sigma_*) \gtrsim \sigma_*$ in Theorem 3.3. We thank the reviewer for pointing out this potentially misleading use of notation. We have clarified in the revision that the following is meant: if $G(3 \sigma_*) \geq c_0 \sigma_*$ for an absolute constant $c_0 > 0$, then there exists another constant $C = C(c_0)$ (i.e., depending on $c$), such that $W_2^2(\mu, \nu) - W_2^2(\mu * N_\sigma, \nu * N_\sigma) \geq C \sigma e^{- \sigma_*^2/\sigma^2}$.

---

> > ### Comment · Reviewer_Wajc · 2022-08-05
> > **Rebuttal**
> >
> > Hi authors,
> >
> > Thank you for answering all of my questions and adjusting manuscript based on my suggestion. I changed my score to 7.

---

### Official Review · Reviewer_KjJi · 2022-07-09

**Rating:** 7
**Confidence:** 3
**Soundness:** 3 good
**Presentation:** 3 good
**Contribution:** 3 good

**Summary:**

This work concerns the asymptotic behaviours of Gaussian-smoothed 2-Wasserstein distance. In particular, the authors provide bounds on the difference between Wasserstein distances of a pair of discrete measures and their smoothened versions when the variances of Gaussian kernels are small. The authors consider two scenarios, when a perfect matching between two discrete measures exists and when it does not. For the former case, they show that the asymptotic gap decays exponentially in some near-zero regions, and linearly otherwise. For the latter case, they show that the gap is linear even in a region around zero.

**Questions:**

Minor Comments:
- In the proof of Theorem 3.1 in the Appendix:
  - Should the truncated kernel be $\mathcal{N}(0, \sigma^2 I) \cdot \mathbf{1}\\{\\|X\\| \le \epsilon_*\\} + (1 - p) \delta_0$?
  - At the end of the proof, for the second inequality, it would be more clear if the authors state which couplings are used to bound the Wasserstein distances.

Questions:
- In the toy experiment, if we slightly modify $\nu$ to $\nu'$ using $k \approx 0$, then there is an unique optimal transport plan between $(\mu, \nu')$, but should the asymptotic behaviors of both pairs $(\mu, \nu')$ and $(\mu, \nu)$ be approximately similar?
- If we look at the Gaussian-smoothened version of a discrete measure as a Gaussian mixture, then does the separability between the means (i.e., the discrete support) play some role in the asymptotic behaviors?
- Do you think that the appearance of the uniqueness of the optimal plan in the study of asymptotic behaviors is related to the use of 2-Wasserstein? For example, if we consider the 1-Wasserstein distance between two sets of points on the real line, and suppose there is some overlapping between the sets, then the optimal plan is not unique, but we can simply remove the overlapped points in both sets without affecting the distance (and consequently the asympotatic behavior) and return to the uniqueness case?

Minor Error:
- In the figures, $p$ should be $k$.

**Limitations:**

The authors have adequately addressed the limitations and potential negative societal impact of their work.

**Strengths And Weaknesses:**

Strengths:
- The paper is well-written and well-presented.
- The addressed questions are of theoretical importance, and the given answers to these questions are novel and complete to the best of my knowledge. The authors covers all possible scenarios, and the tools used for proofs (stronger notions of cyclical monotonicity and implementability, robustness of optimality and their relations) are interesting to me.

Weaknesses:
- In the experiment, the toy example is indeed beneficial for understanding, but I think the authors should also verify their theory in large-scale settings (i.e., large $m, n$ - maybe in the one-dimensional space if the design of points statisyfing the uniqueness of optimal plan is an issue).

---

> ### Author Response · Authors · 2022-08-01
> **Response to reviewer KjJi**
>
> We thank the reviewer for their careful reading and summarizing of our paper and the supplementary proofs. We are glad that the reviewer finds our notions and proofs interesting.
>
> The reviewer suggests that experiments be conducted on a larger scale.
> We did not do so in our original submission since it is quite difficult to see the phenomena we describe when $n$ and $m$ are large.
> For example, in the context of Theorem 4.1, the implicit constant in the linear rate shrinks with $n$, and differentiating between the linear and exponential rates is challenging.
> We therefore chose to present the small-scale experiments present in our submission, since they provided the clearest illustration of our theoretical results.
>
>
> ### Minor Comments.
> We agree with the reviewer that, in the proof of Theorem 3.1, $N(0, \sigma^2 I) \cdot 1(\|X\|\le \epsilon_*) + (1-p) \delta_0$ is a more precise representation of the truncated kernel than in the original paper. In the equation at the end of the proof, the second inequality is yielded by considering the coupling that is the distribution of $\left(X+Z, X+Z\cdot1(\|Z\|\le \epsilon_*)\right)$, where $X$ and $Z$ are independent, $X\sim \mu$ and $Z\sim N(0, \sigma^2 I)$. We have clarified these points in the revised version of the paper. We also thank the reviewer for pointing out the typo in the numerical experiment.
>
> ### Questions
> * In the setting of the toy experiment, the reviewer asked why the asymptotic behavior of the GOT distance between $(\mu, \nu)$ and $(\mu, \nu')$ is so different when $\nu'$ is a small perturbation of $\nu$.
>     As $k$ shrinks, the gap between GOT and the standard $W_2$ distance between $(\mu, \nu')$ converges to the gap between these distance for $(\mu, \nu)$, but there still exists a neighborhood of $0$ for which the error is linear in the case of $(\mu, \nu)$ and exponential in the case of $(\mu, \nu')$.
>     The size of this neighborhood shrinks as $k$ shrinks, a phenomenon which is visible in Figure 2.
>
>
>
> * The reviewer asked if the asymptotic behavior studied in our paper is connected to separability of the means if we look at the source and target distributions $\mu$ and $\nu$ as Gaussian mixtures.
> 	We agree with this perspective---the finite support assumption (which implies positive separation of the means) plays a large role in the theorems we have obtained here.
> 	For example, to go to the opposite extreme where $\mu$ and $\nu$ are absolutely continuous with positive density everywhere (so that we obtain continuous Gaussian mixture), the rates obtained are quite different.
> 	For example, as mentioned in the response to another reviewer, if $\mu = N(0, 1)$ and $\nu = N(0, 4)$, then $|W_2(\mu\ast N_\sigma, \nu\ast N_\sigma) - W_2(\mu, \nu)| \asymp \sigma^2$ for small $\sigma$.
> * The reviewer asked whether our results have strong relations with the use of the $2$-Wasserstein distance. Yes, we believe that the $2$-Wasserstein distance is essential, and that the same asymptotic results would not hold for other Wasserstein distances. Beyond the issue of uniqueness (which, as the reviewer notes, would likely need modification in the case of the $1$-Wasserstein distance), only the $2$-Wasserstein distance possesses a close connection to convex analysis, which is used to derive the characterizations of strong implementability presented in Theorem 2.7 and the bounds of Propositions 2.12 and 2.13.

---

### Author Response · Authors · 2022-08-01
**Thank you for your comments**

We are grateful to the reviewers for their careful reading of the paper. Their feedback has allowed us to prepare a substantially improved version of the manuscript, which we have uploaded along with responses to the reviews.

---

### Meta-Review · Area_Chair_nYq5 · 2022-08-20

**Recommendation:** Accept
**Confidence:** Certain

**Metareview:**

All reviewers are in agreement that the main factors (in particular, the results and their presentation) are above the bar for NeurIPS.  No significant concerns remain following the author response and the discussion period.  I encourage the authors to carefully take into account all of the minor comments when preparing the camera-ready version.

**Award:**

No

---

### Decision · Program_Chairs · 2022-09-14

Accept